# Al₂O₃ and Pt Atomic Layer Deposition for Surface Modification of NiTi Shape Memory Films

**David Vokoun [1,2,\*]**, **Ladislav Klimša [1]**, **Aliaksei Vetushka [1]**, **Jan Duchoň [1]**, **Jan Racek [1]**, **Jan Drahokoupil [1]**, **Jaromír Kopeček [1]**, **Yo-Shane Yu [3]**, **Narmatha Koothan [3]** and **Chi-Chung Kei [3,\*]**

[1] Institute of Physics of the ASCR, Na Slovance 2, 18221 Prague, Czech Republic; klimsa@fzu.cz (L.K.); vetushka@fzu.cz (A.V.); duchon@fzu.cz (J.D.); racek@fzu.cz (J.R.); draho@fzu.cz (J.D.); kopecek@fzu.cz (J.K.)

[2] Department of Biomedical Engineering, Chung Yuan Christian University, 200 Chung Pei Road, Chung Li District, Taoyuan City 32023, Taiwan

[3] Taiwan Instrument Research Institute, National Applied Research Laboratories, Hsinchu 30076, Taiwan; ysyu@narlabs.org.tw (Y.-S.Y.); narmatha26.1992@gmail.com (N.K.)

[\*] Correspondence: vokoun@fzu.cz (D.V.); g893552@narlabs.org.tw (C.-C.K.)

**Abstract:** Pt coatings on NiTi film micro-actuators and/or sensors can add some useful properties, e.g., they may improve the NiTi anticorrosion and thermomechanical characteristics or activate surface properties beneficial for a specific application (e.g., functionalized surfaces for biomedical applications). Pt coatings prepared via atomic layer deposition (ALD) may help reduce cost due to the nanometric thickness. However, no authors have reported preparation of Pt ALD coatings on NiTi films, perhaps due to the challenge of the concurrent NiTi film oxidation during the Pt ALD process. In the present study, Al₂O₃ and Pt ALD coatings were applied to NiTi thin films. The ALD coating properties were studied using electron and atomic force microscopies and X-ray photoelectron spectroscopy (XPS). Potential structural changes of NiTi due to the ALD process were evaluated using electron microscopy and X-ray diffraction. The presented ALD process resulted in well-controllable preparation of Pt nanoparticles on ultrathin Al₂O₃ seed layer and a change of the transformation temperatures of the NiTi films.

**Keywords:** NiTi alloy; atomic layer deposition; thin film; Pt nanoparticles

## 1. Introduction

Atomic layer deposition (ALD) is a kind of chemical vapor deposition (CVD) technique that typically uses two precursor gas reactants sent into the reaction zone in an alternating sequence of precisely controlled pulses. Any excessive amount of unreacted precursors in the deposition chamber is purged with an inert gas. This purging results in the self-limiting character of the surface chemical reactions on the substrate and in this way not more than one precursor monolayer is left on the surface after purging. In ideal cases, the ALD may provide researchers with an excellent thickness control at the Angstrom level, and it forms conformal coating on complex 3D shapes [1–4]. It is worth mentioning that an additional advantage of the ALD is also compositional control demonstrated, e.g., in references [5–9]. Due to low deposition rates (100–300 nm/h [1,10] depending on the size of the deposition chamber and the aspect ratio of the substrate), the ALD is suitable mainly for nanometric coatings. However, successful attempts have been made to increase the deposition rate significantly [11]. When the ALD is applied to coat NiTi alloys, then the deposition temperature (substrate temperature, or specifically, temperature at the NiTi surface) is a concern. High deposition temperature may result in unwanted NiTi grain growth and/or NiTi surface oxidation. Although some ALD processes can take place at low



deposition temperatures [12–14], unfortunately this is not the case for any Pt ALD process applied to NiTi.

Since ALD was introduced as a deposition technique in 1977, a variety of ALD coating types (oxides, nitrides, sulfides, individual chemical elements, etc.) have been prepared. However, for some coatings no suitable precursors have been found, so far. Furthermore, some ALD coatings may be prone to containing organic impurities from precursors [4]. Therefore, for each ALD coating and type of substrate, the ALD process parameters such as growth temperature, pulse duration, precursor pressures, etc., have to be carefully adjusted [15].

The equiatomic NiTi alloy is a thermomechanical transducer material with the ability to convert thermal energy to mechanical energy (and vice versa, when the heat energy is generated in NiTi upon loading in the form of the latent heat of the forward martensitic transformation [16]). This ability is utilized in NiTi actuators whose actuation frequency is usually limited by the slow cooling times of NiTi material [17]. During cooling, the martensitic transformation from B2 to B19′ structure proceeds in NiTi in temperature interval from $M_s$ (martensite start) to $M_f$ (martensite finish) [18]. In order to make the cooling time shorter, NiTi thin films and/or thin wires (both have a high surface-to-volume ratio) have often been employed in NiTi actuators. Besides the relatively fast heat exchange between NiTi films and their surrounding environment, NiTi films may attract interest due to the possibility of incorporating NiTi films in micro-electro-mechanical systems (MEMS) [19–23]. Due to confirmed acceptable biotolerance [24], NiTi films may be also used (with certain limitations) in bio-MEMS. Generally, NiTi and other shape memory alloy (SMA) actuators have high output work (force × stroke) per weight and power density compared to other actuator materials/other types of actuator transducers [22,25], (see also Figure 26 in reference [26]). For instance, microvalves containing NiTi films are capable of generating large work outputs and power densities in the order of 50 mJ/g and 1 W/g, respectively [26]. When pointing out the advantages of using NiTi films in various applications, it is worth mentioning that NiTi sputtered films, unlike NiTi wires, have excellent fatigue resistance [27,28].

The idea to apply thin ALD coatings on NiTi alloys first appeared in the previous study [29]. The authors used an $Al_2O_3$ ALD layer on NiTi to improve the anticorrosive properties of NiTi alloys. Later, Wang et al. [30] presented work on an $Al_2O_3$ ALD layer deposited on a superelastic NiTi carotid stent, where the $Al_2O_3$ layer sufficed for further surface modification with the top heparin layer used as a blood anticoagulant. Anticorrosive and mechanical characteristics of a thin $Al_2O_3$ ALD layer on NiTi were studied by Lin et al. [31]. Piltaver et al. [32] grew $TiO_2/Al_2O_3$ ALD composite films on NiTi and other substrates in order to control the grain size of $TiO_2$. Muralidharan et al. [33] used a $V_2O_5$ ALD layer deposited on the NiTi electrode to control electrochemical reactions in lithium batteries through strains on the $NiTi/V_2O_5$ interface.

In our previous work [29,34,35], $Al_2O_3$, $TiO_2$ and Pt ALD coatings were deposited on NiTi samples and tested for anticorrosive and mechanical properties. Our work related to the ALD coatings on NiTi was motivated by the fact that the ALD is considered an attractive method to grow protective (anticorrosive coatings or diffusion barriers [36]) and biocompatible coatings due to excellent coating conformity, sound adhesion and low-temperature compatibility. Besides that, ALD coatings can also serve when forming 3D nanostructures [37] as a part of MEMS.

ALD Pt coating has some advantages over $Al_2O_3$ and $TiO_2$ ALD coatings as follows. First of all, Pt coating has the potential to withstand deformations larger than the deformation allowable for ceramic coatings. Pt coating may increase radio-opacity of NiTi stents (in order to increase radio-opacity of biomedical grade NiTi wires, NiTi–Pt wires with a Pt core were proposed in 2004 and subsequently have been produced on a commercial basis [38]). Furthermore, it can form an interlayer for further surface modifications. Pt coating increases surface electrical conductivity and surface energy [39,40]. In addition to Pt prepared as a continuous layer, Pt nanoparticles (NPs) can also be prepared on various surfaces using the ALD process [41]. Pt NPs on $TiO_2$ surfaces can have several applications, such as an efficient water-splitter [42] or an efficient electro-catalyst for methanol oxidation [41].

The ALD process for Pt on NiTi is not straightforward. The usually used precursors are (trimethyl)-methylcyclopentadienyl-platinum (*IV*) (MeCpPtMe$_3$) [43–46] or Pt(acac)$_2$ (acac = acetylacetonate) [47,48]. Precursor dimethyl(*N*,*N*-dimethyl-3-butene-1-amine-*N*)platinum (DDAP, C$_8$H$_{19}$NPt) has also been previously explored in some detail [49].

The co-reacting precursor is usually oxygen or dry air. The temperature window in the case of the MeCpPtMe$_3$ precursor is 200–300 °C [47]. The Pt ALD deposition rate is low on TiO$_2$ [50] and TiO$_2$ is regularly present on the NiTi alloy surface because of NiTi oxidation, even at room temperature [51]. In addition, the Pt ALD process (about 100 ALD cycles) on TiO$_2$ results in a low spreading density of Pt NPs [50]. The deposition rate of Pt on Al$_2$O$_3$ is superior to the rate of Pt on TiO$_2$. Therefore, it is suggested that an Al$_2$O$_3$ underlayer may facilitate initial ALD growth better than the original TiO$_2$ on NiTi. It is worth mentioning that the authors of reference [39] (also reference [52]) achieved a thin ALD Pt compact layer by depositing an additional W ALD layer on Al$_2$O$_3$ as an adhesion layer.

In the present study, the growth of Pt coating on an Al$_2$O$_3$ underlayer is evaluated using various techniques such as transmission electron microscopy (TEM), X-ray photoelectron spectroscopy (XPS) and atomic force microscope (AFM). The thermal influence of the ALD processes on the grain size and transformation temperatures of NiTi films is examined as well.

## 2. Materials and Methods

### 2.1. Samples

Transforming NiTi films with a thickness of about 1 µm deposited on Si substrates (5 × 10 mm$^2$) were obtained from Acquandas GmbH. In reference [53], experimental work showing determination of the transformation temperatures of the as-purchased NiTi films was described. Before the ALD, the as-purchased NiTi samples were cleaned in ethanol and deionized water and finally dried using N$_2$ gas.

### 2.2. ALD

The deposition was performed in a home-built ALD reactor assembled (including all accessory parts) in Taiwan Instrument Research Institute (TIRI). The precursors used for the Al$_2$O$_3$ (or Pt) ALD process were trimethyl-aluminum (TMA) and water vapor (or (methylcyclopentadienyl)trimethyl-platinum (MeCpPtMe$_3$), a product of Strem Chemicals and dry air as a source of O$_2$). First, Al$_2$O$_3$/NiTi/Si samples with 10 Al$_2$O$_3$ ALD cycles and subsequently Pt/Al$_2$O$_3$/NiTi/Si samples with 100 and 200 Pt ALD cycles were prepared. The substrate temperature during the Al$_2$O$_3$ (or Pt) ALD process was 100 °C (or 300 °C). N$_2$ gas was exclusively used for purging. The base pressure was about 3.33 Pa. The typical cycle times for Al$_2$O$_3$ (or Pt) ALD growth experiment were as follows: 0.2 s TMA pulse, 10 s N$_2$ purge, 0.2 s water vapor pulse, and 10 s N$_2$ purge (or 0.5 s MeCpPtMe$_3$ pulse, 10 s N$_2$ purge, 0.5 s air pulse, and 10 s N$_2$ purge). Hence, the Pt ALD process took about 35 min (Pt 100 ALD cycles) and 70 min (Pt 200 ALD cycles). In the initial Pt ALD cycle, hydrogen plasma was used (power output 300 W, deposition temperature 200 °C, H$_2$ pulse duration 5 s) instead of dry air [39]. Finally, Table 1 lists all the used film samples with their names.

**Table 1.** A list of the film samples and their short names used in the further text.

| Sample | Name |
|---|---|
| NiTi/Si | NiTi |
| Al$_2$O$_3$_ALD_10_cycles/NiTi/Si | Alu10 |
| Pt_100_ALD_cycles/Al$_2$O$_3$_ALD_10_cycles /NiTi/Si | Pt100 |
| Pt_200_ALD_cycles/Al$_2$O$_3$_ALD_10_cycles /NiTi/Si | Pt200 |

### 2.3. Microscopic Observations

Surface observations of samples Pt100 and Pt200 were made using a scanning electron microscope (SEM) TESCAN, FERA3 GM (Brno, Czech Republic). The NiTi grain size distributions in samples NiTi and Pt100 were obtained based on electron backscattering diffraction (EBSD) method using the EDAX system (EDAX, Mahwah, NJ, USA) with DigiView IV camera. It was possible to get the EBSD signal in NiTi and Pt100 samples, whereas no signal was obtained in sample Pt200, because the layer covering the NiTi layer was too thick.

A thin cross-sectional lamella was cut out from sample Pt100 (using a Ga focused ion beam in an FEI Quanta 3D Dual-Beam SEM, FEI, Hillsboro, OR, USA) and observed with a Fei Tecnai F20 field emission gun transmission electron microscope (TEM, FEI, Hillsboro, OR, USA) operated at 200 kV. Selected area electron diffraction and energy dispersive X-ray spectroscopy (EDS) were performed to identify phases and chemical elements, respectively.

Surface morphology and local conductivity of samples NiTi, Alu10 and Pt100 were investigated using Dimension Icon AFM (Bruker, Billerica, MA, USA) with an SCM-PTSI platinum silicide coated probe with a tip radius of ca. 15 nm. All the measurements were performed at room temperature under ambient conditions. The topography images were measured in PeakForce mode whereas the maps of local current were measured either in contact mode or PeakForce mode (PeakForce mode was used in the case of the simultaneous measurement of topography and local conductivity). The velocity of scanning varied from 250 to 1000 nm/s. The cantilever deflection was detected by a red laser diode (685 nm).

### 2.4. XPS and XRD (X-ray Diffraction) Measurements

The XPS spectra were obtained on a Kratos Axis Supra spectrometer (Kratos Analytical, Manchester, UK) with monochromatic Al K$\alpha$ (1486.6 eV) X-ray radiation. The pressure in the analysis chamber was maintained near $10^{-9}$ Pa. For surface cleaning and depth profiling, a multi-mode Ar gas cluster ion source (GCIS) was used with an acceleration voltage of 5 keV and the beam current about 5.8 nA. The scanned area, dwell time in one step and number of steps were $2 \times 2$ mm$^2$, 10 min and 6 steps, respectively. The mean cluster size was around 2000 (or 1000) argon atoms/cluster in the first four steps (or in the last two steps). It was estimated that 10 min of Ar ion bombardment corresponds to the depth ranging from few to several nm in the ALD layers. The signal was taken from a circular area with a diameter of 100 µm on sample Pt200. The XPS input data were compiled and analyzed using Gaussian–Lorentzian functions and Shirley background implemented in CasaXPS software (version 2.3) [54].

In the present study, the $M_s$ transformation temperatures of samples Pt100 and Pt200 were determined using XRD in order to find out any effect of the Pt and Al$_2$O$_3$ ALD processes applied on NiTi/Si samples. The XRD signal was registered from the film surface using an X-ray diffractometer (X'Pert PRO from PANalytical, Malvern, UK) with a thermal chamber. Before any XRD measurement, either the NiTi film was first heated above the $A_f$ (austenite finish) temperature and then cooled down to a test temperature. The XRD patterns were registered at various test temperatures, which were kept constant throughout each XRD measurement. The X-ray source was Co K$\alpha$ radiation ($\lambda_{K\alpha1}$ = 0.178901 nm, $\lambda_{K\alpha2}$ = 0.17929 nm). The $M_s$ temperature was determined based on the change of the full width at half maximum of (110)$_{B2}$ peak, using the fact that during the martensitic transformation, peak (110)$_{B2}$ splits into two, or more neighboring peaks. Determination of the $M_s$ temperature (of the as-purchased NiTi film) was shown in reference [53].

## 3. Results

### 3.1. Microscopic Observations

The microscopy results are expected to reveal (i) whether the Al$_2$O$_3$ ALD layer in samples Alu10, Pt100 and Pt200 is continuous and (ii) whether Pt in samples Pt100 and Pt200 forms nanoparticles (NPs)

or a continuous layer and, finally, (iii) whether the grain size distribution in the NiTi layer changes after the $Al_2O_3$ and Pt ALD processes.

Figure 1a,b shows tiny Pt NPs, at the limit of SEM resolution, on the surfaces of samples Pt100 (a) and Pt200 (b). The areal density and mean size of the Pt NPs in Pt200 (Figure 1b) were higher than those in Pt100 (Figure 1a). The sizes of the Pt NPs in Figure 1a,b were 9–17 and 12–20 nm, respectively.

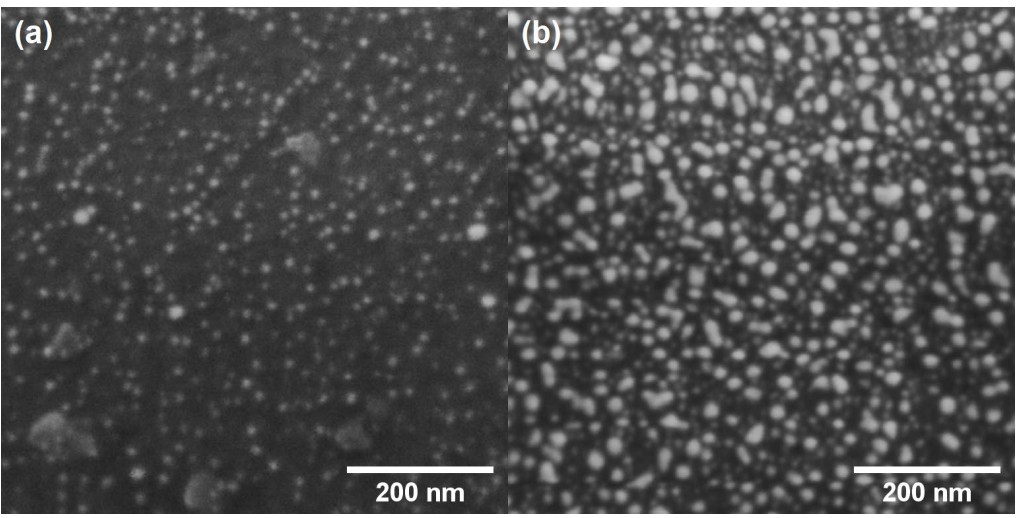

**Figure 1.** The SEM micrographs of the surfaces of samples Pt100 (**a**) and Pt200 (**b**).

After the $Al_2O_3$ and Pt ALD processes, the grain size and grain distribution in the NiTi layer changed (due to the Pt deposition temperature up to 300 °C) as shown in Figure 2. Figure 2 compares grain distributions in the NiTi layers of samples NiTi (before the ALD depositions) and Pt100 (after the ALD depositions).

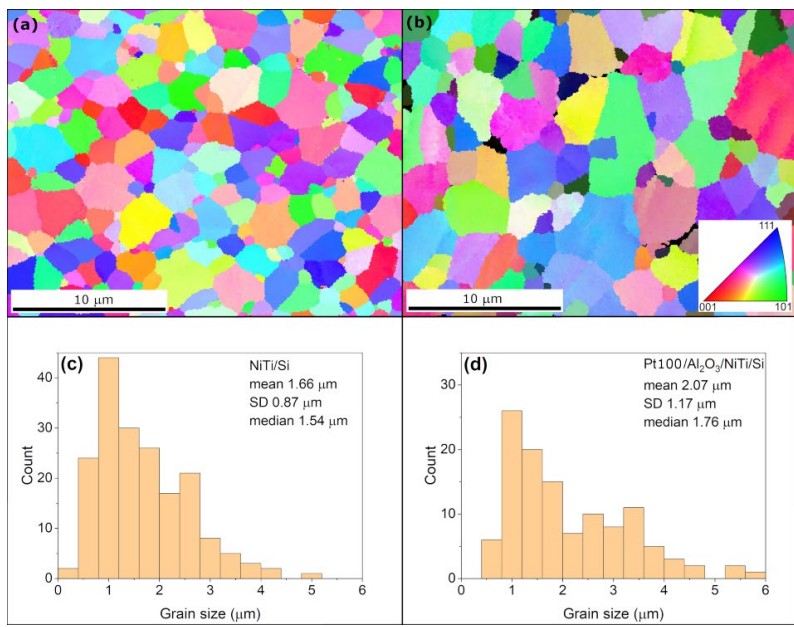

**Figure 2.** The color electron backscattering diffraction (EBSD) maps representing the orientation of the grains through inverse pole figure (IPF) color-coding in selected surface areas of NiTi (**a**) and Pt100 (**b**) samples and the corresponding grain size distributions in NiTi (**c**) and Pt100 (**d**) samples. The color code for IPF orientation is given in the inset of part (**b**).

Figure 3a,b shows the TEM cross-sectional micrographs of sample Pt100, with low (a) and high magnification (b). The Si substrate in the low-magnification image is seen in a light color whereas the W layer (deposited via gas injection system (GIS) to manipulate Pt100 lamella) is seen in a dark color. The high-angle annular dark-field scanning transmission electron microscopy (HAADF STEM) image shows the upper layers in the cross-section (Figure 3c). The path of the line scan with a fixed point denoted as point 1 are shown in the image whereas the results of the EDS analysis along the line are shown in Figure 3d. The presence of Ga, Cu and W elements originates from Ga focused ion beam used to cut out the examined lamella, a Cu ring used as a support for the lamella and GIS material, respectively. The electron diffraction pattern taken from the SAED (selected area electron diffraction) inside of a NiTi grain is shown in Figure 3e. The pattern corresponds to the R-phase (the rhombohedral structure); the zone axis was [−121].

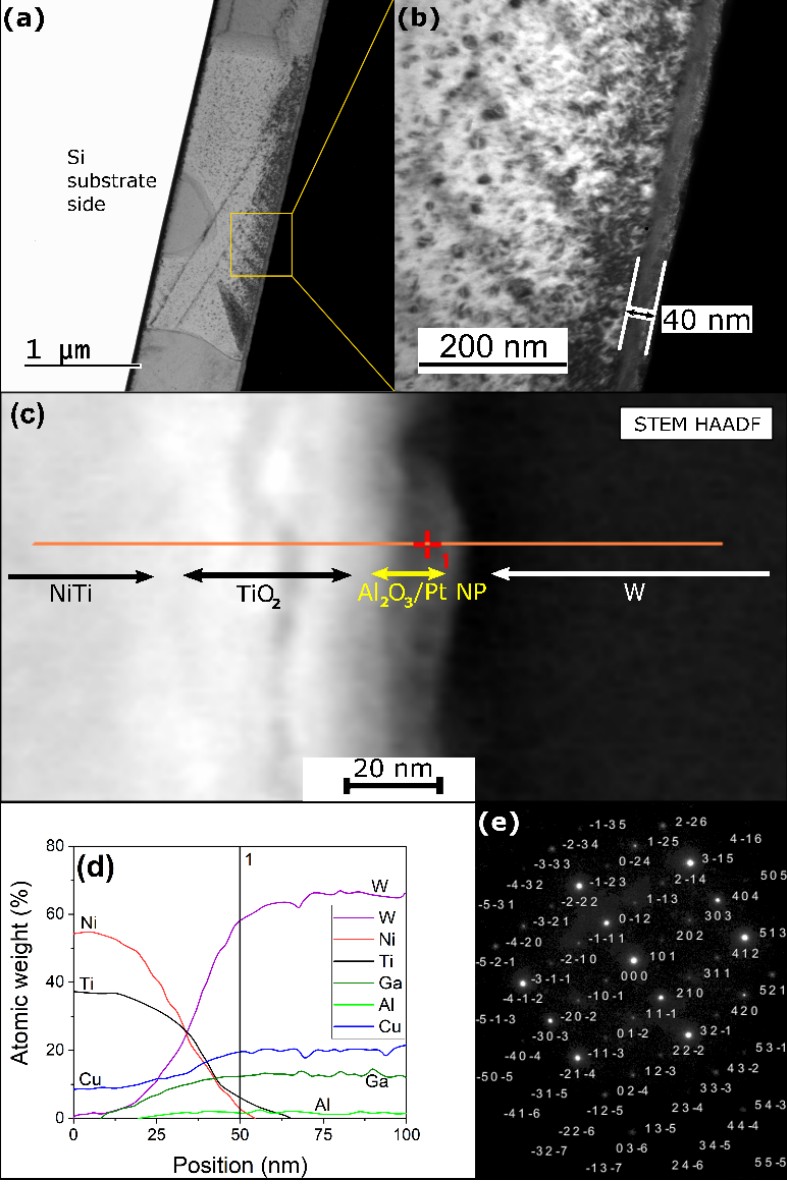

**Figure 3.** The TEM cross-sectional micrograph of sample Pt100 taken in the bright field (BF) TEM mode (**a**,**b**), high-angle annular dark-field scanning transmission electron microscopy (HAADF STEM) mode (**c**), EDS line scans close to the NiTi top surface, however the tungsten layer is not a part of the sample (**d**), and the diffraction pattern (corresponding to the R-phase) taken from the SAED (selected area electron diffraction) inside of a NiTi grain (**e**).

Figure 4a–c shows AFM topography images of samples NiTi (a), Alu10 (b) and Pt100 (c). The surface patterns in Figure 4a,b (belonging to samples NiTi and Alu10) are regular. The round objects in Figure 4a,b are 10–30 nm in size. Figure 4c shows surface of sample Pt100 containing two kinds of round objects—the small object (with high occurrence) and the large objects (with low occurrence) with diameters 10–17 and 30–50 nm, respectively. The cross section analysis in Figure 4d corresponds to the white line in Figure 4a. The maps of local current obtained by conductive AFM (C-AFM) for samples NiTi (a), Alu10 (b) and Pt100 (c) are shown in Figure 5a–c. The scan direction is from the top downward. Figure 5a shows high currents about −40 pA at the top and currents of few pA at the bottom. A similar distribution of local currents is shown in Figure 5b but with much lower values of local currents. Figure 5c shows some conductive round objects with diameters ranging 10–17 nm. The cross section analysis in Figure 5d corresponds to the white line in Figure 5a.

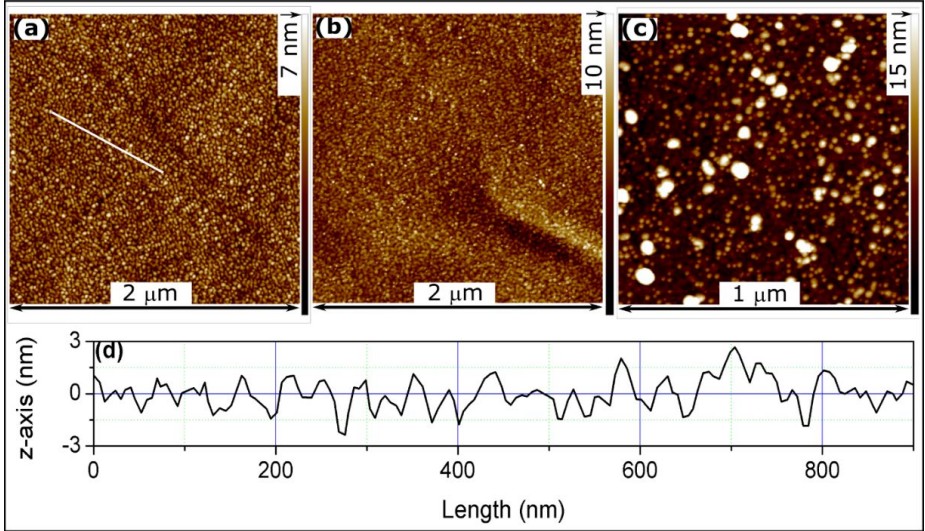

**Figure 4.** The atomic force microscope (AFM) topography images of NiTi sample's surface (forces 500 pN) (**a**), Alu10 sample's surface (forces 500 pN) (**b**) and Pt100 sample's surface (the maximum interaction force was 5 nN; measured simultaneously with local conductivity (Figure 5c)) (**c**). The white line in subfigure (**a**) indicates positions of the cross section analysis (**d**).

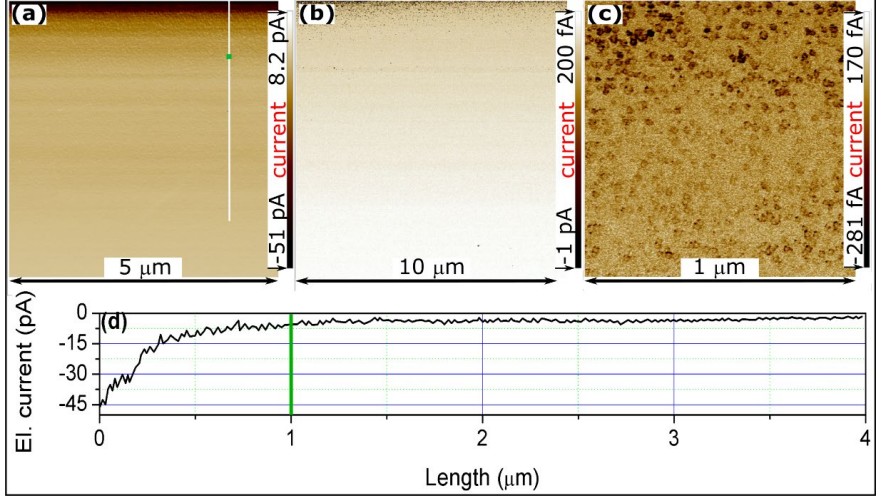

**Figure 5.** The maps of local current obtained by C-AFM for samples NiTi (forces 100 nN, sample bias −8 V) (**a**), Alu10 (forces 100 nN, sample bias −4 V) (**b**) and Pt100 (the maximum interaction force was 5 nN, sample bias −5V, measured simultaneously with topography (Figure 4c)) (**c**). The white line in subfigure (**a**) indicates positions of the cross section analysis (**d**).

## 3.2. XPS and XRD Measurements

The diagrams of measured and fitted XPS spectra expressed as counts per second (CPS) versus binding energy (BE) for several individual elements after various $Ar_{n+}$ etching times are shown in Figures 6–8.

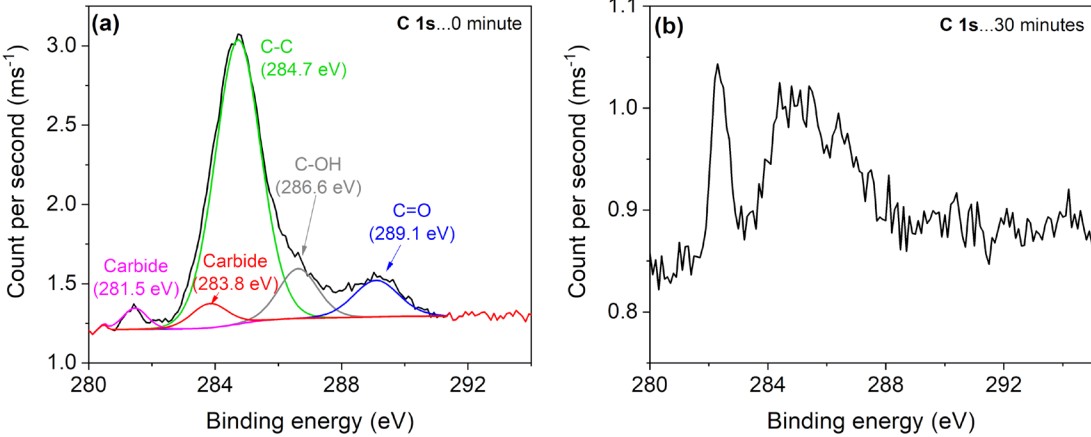

**Figure 6.** The measured and fitted XPS spectra attributed to C before etching (**a**) and the measured XPS spectrum after 30 min of $Ar_n^+$ etching (**b**).

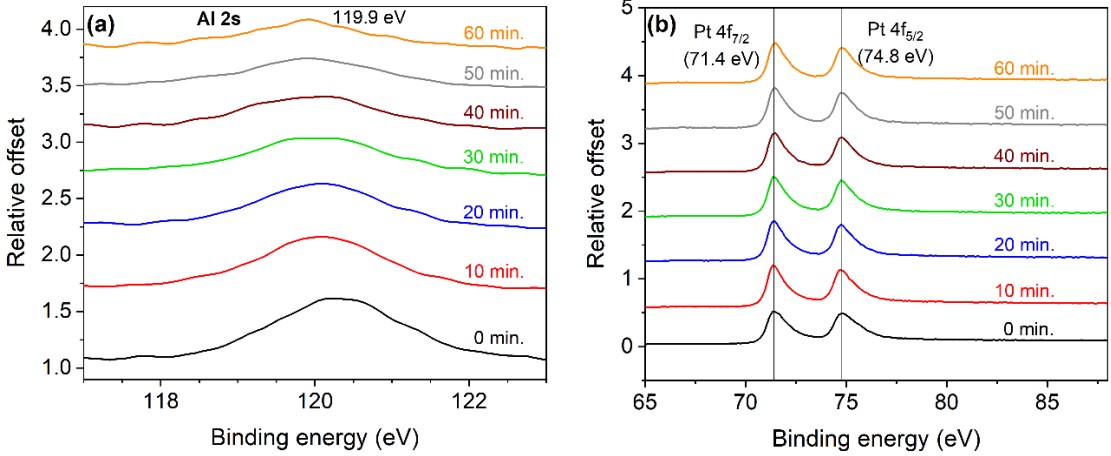

**Figure 7.** The measured XPS spectra attributed to Al (**a**) and Pt (**b**) after various etching times.

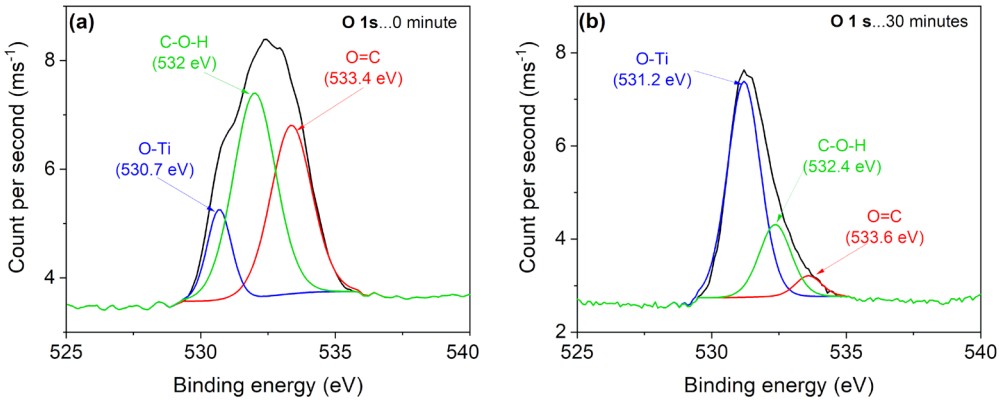

**Figure 8.** The measured and fitted XPS spectra attributed to O before etching (**a**) and after 30 min of $Ar_n^+$ etching (**b**).

Specifically, Figure 6a,b shows the diagrams of CPS versus BE attributed to C before $Ar_n^+$ etching (0 min) and after 30 min of $Ar_n^+$ etching, respectively. The diagrams of CPS versus BE, attributed to C, after etching times 10, 20, 40, 50, and 60 min (not shown here) are similar to the diagram in Figure 6b. Figure 6b does not contain any fitted XPS spectra because the measured XPS data are too noisy due to a relatively small amount of C, in various forms. Figure 7a,b shows the diagrams of CPS (relative offset) versus BE attributed to Al and Pt after various $Ar_n^+$ etching times, respectively. Figure 8a,b shows the diagrams of CPS versus BE attributed to O before $Ar_n^+$ etching (0 min) and after 30 min of $Ar_n^+$ etching, respectively. Again, the diagrams of CPS versus BE, attributed to O, after the other etching times (not shown here) are similar to the diagram in Figure 8b. Atomic percentages of the elements at the surface of sample Pt200 determined from the XPS after various times of $Ar_n^+$ etch are summarized in Table 2.

**Table 2.** Atomic percentages of the elements at the surface of sample Pt200 determined from the XPS after various times of $Ar_n^+$ etch.

| Etch (min) | Pt (at.%) | Al (at.%) | C (at.%) | Ti (at.%) | O (at.%) | Ni (at.%) |
|---|---|---|---|---|---|---|
| 0 | 4.75 | 8.47 | 37.99 | 4.65 | 44.15 | 0.00 |
| 10 | 7.73 | 9.23 | 10.55 | 17.34 | 53.95 | 1.19 |
| 20 | 9.05 | 9.20 | 9.13 | 18.53 | 52.17 | 1.92 |
| 30 | 9.21 | 8.54 | 9.80 | 19.33 | 50.85 | 2.27 |
| 40 | 9.71 | 8.19 | 9.44 | 19.79 | 49.88 | 2.99 |
| 50 | 10.58 | 7.14 | 8.84 | 20.84 | 48.38 | 4.21 |
| 60 | 11.14 | 7.17 | 7.23 | 21.36 | 47.34 | 5.77 |

The transformation temperatures of sample NiTi and the $M_s$ temperatures of samples Pt100 and Pt200 are summarized in Table 3. At room temperature each sample may contain both phases, austenite and martensite with crystal structures B19′ and B2, respectively. The diffractograms for the as-purchased NiTi film measured at various temperatures are shown in reference [53]. The diffractograms for samples Pt100 and Pt200 measured in the $(110)_{B2}$ neighborhood at various temperatures between 25 and 90 °C are shown in Figure 9. Diffraction peaks $(110)_{B2}$ are not shown for all the test temperatures but only for few selected temperatures in order to avoid accumulating too much data. The $M_s$ temperatures for samples Pt100 and Pt200 are determined from the diagrams of FWHM (full width at half maximum) versus temperature (see the insets of Figure 9)

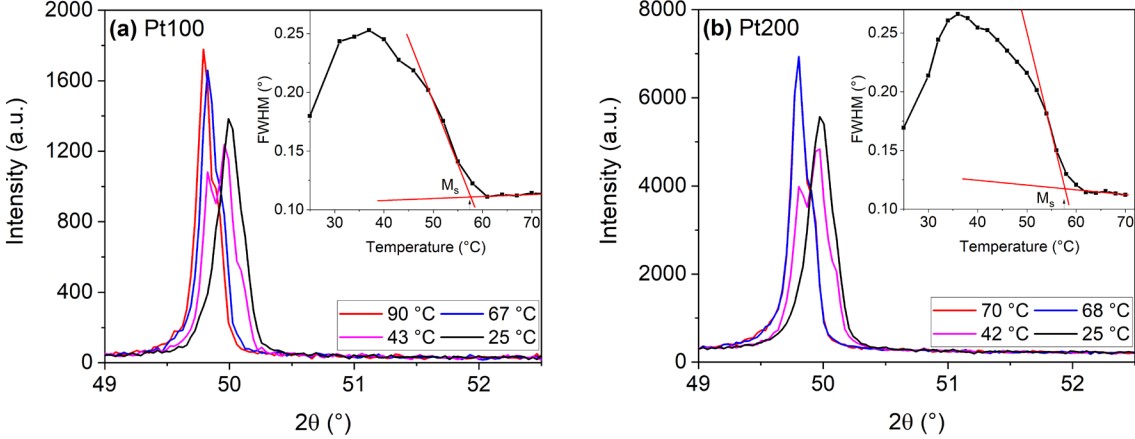

**Figure 9.** The diffractograms for samples Pt100 (**a**) and Pt200 (**b**) measured in the $(110)_{B2}$ neighborhood at various temperatures between 25 and 90 °C. The insets show the corresponding FWHM (full width at half maximum) of the $(110)_{B2}$ peaks in dependence on temperature.

**Table 3.** The transformation temperatures of the NiTi sample determined with AFM, from XRD patterns and shifts of resonant frequency peaks (results taken from reference [53]) and the $M_s$ temperatures attributed to samples Pt100 and Pt200 and determined from the XRD patterns registered at various temperatures.

| Sample | Transformation Temperature | AFM | XRD | Resonant Ultrasound Spectroscopy |
|--------|---------------------------|-----|-----|----------------------------------|
| NiTi | $M_s$ (°C) | 43 | 43 | 44 |
| NiTi | $M_f$ (°C) | - | - | 17 |
| NiTi | $A_s$ (°C) | - | - | 33 |
| NiTi | $A_f$ (°C) | - | - | 75 |
| Pt100 | $M_s$ (°C) | - | 57 | - |
| Pt200 | $M_s$ (°C) | - | 57 | - |

## 4. Discussion

The discussion is separated into three parts related to (i) properties of Pt coating on samples Pt100 and Pt200, (ii) properties of $Al_2O_3$ coating on samples Alu10, Pt100, Pt200 and (iii) the effect of the Pt and $Al_2O_3$ ALD processes on the properties of the NiTi layer.

### 4.1. Pt Phase

Pt in samples Pt100 and Pt200 did not form a continuous layer but NPs. The sizes of the Pt NPs on the surfaces of samples Pt100 and Pt200 were 9–17 nm (determined from Figures Figures 1a and 4c) and 12–21 nm (determined from Figure 1b), respectively. The areal densities of Pt NPs in samples Pt100 and Pt200 were about 800–1000 NPs per 1 $\mu m^2$ (determined from Figures Figures 1a and 4c) and about 1700 NPs per 1 $\mu m^2$ (determined from Figure 1b), respectively. As for the Pt NP size range and the Pt NP areal density on sample Pt100, there is a fair agreement between the results obtained from the SEM (Figure 1) and AFM images (Figure 5c). Due to the relatively low density of Pt NPs on sample Pt100, the EDS elemental analysis (in the chamber of TEM) along a line across the top surface did not show any presence of Pt (Figure 3d). There is no clear interface between the individual deposited layers in Figure 3c. Therefore, the locations of the individual layers were hinted at in Figure 3c. The Pt NPs are conductive (Figure 5c) and do not contain any significant amount of PtO or $PtO_2$ phases as implied from the values of XPS binding energies corresponding to the peaks in Figure 7b. Specifically, according to reference [55], binding energies corresponding to Pt $4f_{7/2}$ for Pt, PtO and $PtO_2$ are 71.1 ± 0.3, 72.2 ± 0.3, and 74.2 ± 0.3 eV, respectively. The XPS binding energy registered from sample Pt200 and corresponding to Pt $4f_{7/2}$ was 71.4 eV (Figure 7b), a value close to 71.1 eV (a BE value characteristic for Pt) measured in [55]. Some organic residua may remain on the surfaces of samples Pt100 and Pt200 as seen from Figure 6a. Specifically, Figure 6a shows carbon peaks with BE corresponding to C–C, C–OH and C=O bounds (the bounds characteristic for organic material). In addition, Table 2 shows that the non-etched surface of Pt200 contains about 38 at.% of carbon. After the Arn+ etch, the content of carbon significantly dropped to values around 10 at.%. The origin of the organic residua is likely in the Pt organic precursor. However, the distinct advantage of the Pt ALD process is the controllability of Pt NP size and the areal density of Pt NPs by controlling the number of Pt ALD cycles (as shown also in reference [56]).

### 4.2. $Al_2O_3$ Phase

In general, $Al_2O_3$ ALD processes and properties of $Al_2O_3$ ALD layers are discussed in references [57–59]. Thin $Al_2O_3$ ALD layers grown below 600 °C are amorphous regardless of the type of the used substrate [58]. Hence, the $Al_2O_3$ coating in the present study was amorphous. In our previous study [29], an almost identical $Al_2O_3$ ALD process applied to NiTi plate samples resulted in the deposition rate of the 0.1 nm per ALD cycle. Therefore, it is assumed that if the $Al_2O_3$ coating in samples Alu10, Pt100 and Pt200 is continuous then its thickness is about 1 nm.

After comparing Figure 5a,b one can see that the surface of sample NiTi is much more conductive than the surface of sample Alu10. Since the map of the local current in Figure 5b is quite uniform (no

islands are obvious), it can be deduced that the $Al_2O_3$ coating (10 ALD cycles) in sample Alu10 forms an insulating and continuous layer. It is assumed that the $Al_2O_3$ layers in samples Pt100 and Pt200 are also insulating and continuous. Some other properties of the $Al_2O_3$ layer can be found from the XPS measurement (Figure 7a). Figure 7a shows that the individual Al 2s peaks shift with the changing $Ar_n^+$ etch time. This effect may be an artifact from the Ar-etch.

Here, the $Al_2O_3$ layer has a double function: (i) $Al_2O_3$ as a barrier of further oxidation of NiTi at higher temperatures (the Pt ALD process requires high temperatures on the samples' surfaces, about 300 °C) and (ii) $Al_2O_3$ as a layer allowing the Pt growth rate to be higher than that on $TiO_2$.

Apart from the properties of the Pt and $Al_2O_3$ ALD coatings, it is worth mentioning the absence of Ni (in any form) on the top surface of sample Pt200 (Table 2). Generally, Ni from a NiTi alloy may be released in a corrosive environment in the form of Ni ions, which is of concern when the NiTi alloy is used as an implant in the human body. Furthermore, as the content of $TiO_2$ is concerned, Table 2 and Figure 8a,b show the difference between the surface of sample Pt200 before and after the $Ar_n^+$ etch. There is less $TiO_2$ before the $Ar_n^+$ etch than after the etch. The presence of $TiO_2$ is due to the oxidation of NiTi layer at room temperature and during the Pt and $Al_2O_3$ ALD processes.

### 4.3. The Effect of the Pt and $Al_2O_3$ ALD Processes on Properties of the NiTi Layer

The deposition temperatures of the Pt and $Al_2O_3$ ALD processes were 300 and 100 °C, respectively. The $Al_2O_3$ ALD process was relatively short (only 10 ALD cycles) with the relatively low deposition temperature. Therefore, the Pt ALD process predominantly influenced the properties of the NiTi layer in samples Pt100 and Pt200. Properties of NiTi alloys and other SMAs depend to a large extent on the following factors: (i) chemical composition, (ii) grain size, (iii) texture, (iv) internal stresses, etc. Each factor (of factors (i)–(iv)) may change due to annealing of an SMA in question. The effects of factors (i)–(iv) may be inter-related and/or unevenly distributed in an SMA sample and it may be hard to discriminate contributions of the individual factors as, e.g., in the case of Fe-30 at.% Pd SMA melt-spun ribbons [60]. Since the NiTi layers in samples Pt100 and Pt200 were firmly attached to the substrate it was not easy to study mechanical properties of the NiTi layers but it was convenient to determine the $M_s$ transformation temperatures (an important parameter of SMAs). Table 3 indicates an increase of the $M_s$ temperature of the NiTi layers in samples Pt100 and Pt200 by 14 °C, due mainly to the Pt ALD process. Both the Pt (100 ALD cycles) and Pt (200 ALD cycles) processes resulted in the identical shift of the $M_s$ temperature implying that in samples Pt100 and Pt200, the exposure time (about 35 and 70 min) did not matter as much as the exposure temperature (300 °C). In our previous work, a similar effect on the transformation temperatures was observed when exposing a thin NiTi plate to a polymerization temperature of 200 °C for 2 h during manufacturing the NiTi-PI composite. The PI polymerization process in the NiTi-PI composite resulted in the drop of the $M_s$ temperature by about 3 °C [61]. In both cases (NiTi-PI and Pt100/200 samples) it is hard to find the exact cause of the $M_s$ shift. In the case of Pt100/200 samples, oxidation of NiTi, precipitation ($Ni_4Ti_3$), stress due to the difference in the coefficients of thermal expansion of the substrate and the NiTi alloy might play a certain role in the increase of $M_s$.

As for the grain size distribution change due to the Pt and $Al_2O_3$ ALD processes, there was a minor increase of NiTi grains in sample Pt100 (Figure 2a–d). However, the examined areas in samples NiTi and Pt100 were too small to be statistically significant. The main purpose of the EBSD measurements was to make sure that the exposure of the NiTi film to temperatures as high as 300 °C did not result in a remarkable change of the grain size distribution.

## 5. Conclusions

The $Al_2O_3$ ALD process consisting of ten ALD cycles and applied to NiTi films results in a continuous and insulating layer. The following Pt ALD coating is not continuous but forms NPs with sizes 9–17 nm (Pt100) and 12–21 nm (Pt200). The areal densities of Pt NPs are about 800–1000 NPs per 1 $\mu m^2$ (Pt100) and about 1700 NPs per 1 $\mu m^2$ (Pt200). The Pt NPs are conductive and do not contain any

significant amount of PtO or $PtO_2$ phases. The distinct advantage of the Pt ALD is the controllability of Pt NP size and the areal density of Pt NPs by controlling the number of Pt ALD cycles.

The $Al_2O_3$ and Pt ALD processes cause non-significant grain size growth and an increase of the $M_s$ transformation temperature by 14 °C.

**Author Contributions:** Conceptualization, D.V. and C.-C.K.; methodology, L.K., A.V., J.D. (Jan Duchoň), J.D. (Jan Drahokoupil), Y.-S.Y., N.K., and C.-C.K.; formal analysis, A.V., J.D. (Jan Duchoň), and J.D. (Jan Drahokoupil); investigation, J.D. (Jan Drahokoupil) and D.V.; data curation, J.R.; writing—original draft preparation, D.V.; writing—review and editing, D.V.; visualization, D.V., A.V., J.R., and L.K.; Supervision, J.K. All authors have read and agreed to the published version of the manuscript.

**Funding:** This research was funded by the Czech Academy of Sciences and the Ministry of Science and Technology, R.O.C. within a Czech-Taiwanese Joint Research Project No. MOST-20–11. Furthermore, the work was supported by Operational Program Research, Development and Education financed by European Structural and Investment Funds and the Czech Ministry of Education, Youth and Sports (Project No. SOLID21-CZ.02.1.01/0.0/0.0/16_019/0000760).

**Acknowledgments:** We thank Jiříček from Institute of Physics, ASCR for carrying out the XPS measurement.

**Conflicts of Interest:** The authors declare no conflict of interest. The funders had no role in the design of the study; in the collection, analyses or interpretation of data; in the writing of the manuscript, or in the decision to publish the results.

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
