# Peer review of "Al2O3 and Pt Atomic Layer Deposition for Surface Modification of NiTi Shape Memory Films"

_coatings, doi:10.3390/coatings10080746_

Round 1

Reviewer 1 Report

The quality of the presentation in the paper by Vokoun et al., ID coatings-876255 is rather poor. Some problems may just be related to semantics, but the readability definitely suffers from that. The questionable issues can be described as follows:

  1. The abstract is not written very appropriately. It is, at first, too detailed and, secondly, does not meet the required style. An abstract to a scientific paper should, in principle, present in a most concise manner both goals and results of the particular study. However, the authors have, for instance, written in the Abstract, that „NiTi films were kept at 300 deg. C for about 45 minutes (Pt 100 ALD cycles) and 90 minutes (Pt 200 ALD cycles) resulting in a moderate NiTi grain growth (Pt 100 ALD cycles) and an increase of the Ms transformation temperature by 14°C (both Pt 100 and 200 ALD cycles).“ At first, the exact amounts of the minutes and ALD cycles are not to be considered as possible data. These data can possibly not be generalized, they are reactor-dependent.
  2. What is this MS? This appears in Section 2.4 in row 155, without any explanation and opening the meaning of Ms for a reader. At the same time, it seems that MS (whatever it may be) is the only physical quantity which can be linked to shape memory properties of NiTi substrates, is this correct?
  3. It is said that the MS transformation temperatures were determined using XRD. However, there are no diffractograms presented in this manuscript. Instead, there are lot of data obtained by AFM and XPS, which seems to be relatively less important compared to XRD.
  4. What is the acronym RUS in the Table 3 ?
  5. The authors have written in rows 40-41, that „When the ALD is applied to coat NiTi alloys, then the deposition temperature (substrate temperature, or specifically, temperature at the NiTi surface) is often a concern.“ What is meant by the word „often“ here?
  6. The authors have written in rows 43-44, that „an advantage of the ALD process is low temperature deposition for a large number of precursors [10-12]“ Do these three references 10-12 really describe exploitation of a „large“ number of precursors?
  7. The authors tell a reader, within rows 45-50, about the importance to use different ALD coatings using different chemistries and refer to the importance of chemical precursors and impurities arising from such chemicals. All this can be true, but how does this relate to the present manuscript and its goals?
  8. In rows 93-94, the authors write that „The ALD process for Pt on NiTi is not straightforward. The standard precursors are (trimethyl)-methylcyclopentadienyl-platinum(IV) (MeCpPtMe3) [40-43] or Pt(acac)2 (acac=acetylacetonate) [44,45]“. Yes, the Pt deposition process is not straigtforward on any substrate, and also there are no „standard“ precursors for this process. Precursors chemicals applied in any paper can not yet be called standard precursors. Besides, the authors refer to studies devoted to the platinum growth on TiO2 and say in the row 99 that, therefore, boosting of growth of Pt on NiTi is needed. Why „therefore“?
  9. In the Section 2.2, it seems that the reactor type and conditions are not revealed for a reader.
  10. In the section 4.2, the authors tell, that the shift of the Al 2s peaks in XPS spectra is likely due to the changing Al/O ratio with the changing number of the ALD cycles. How can that be possible, if the number of Al2O3 growth cycles was fixed at 10?
  11. It is not exactly understood, why the authors have organised TEM on Pt100 sample (Fig. 3). There is nothing in this Figure, which could be distinguished and recognized as traces of platinum (dots). Also the 1 nm thick Al2O3 layer can not be visualized. Instead, there are EDS profile from supporting materials (Ni, Ti, Cu, and W) and diffraction pattern from the substrate, which can both be considered as natural parts of the sample, even trivial data. Besides, where did the TiO2 layer visible in panel c) come out? The TiO2 was not included in the description of sample Pt100 in Table 1. In fact the whole Figure 3 looks somewhat unnecessary.
  12. In the end, it became obvious, that after applying the revealed low amount of Pt growth cycles, the authors have still achieved a distribution of the Pt nanodots. Why did the authors not attempt the growth of continuous Pt films just by increasing the amount of deposition cycles? What was expected - how could a distribution of the discontinuous Pt dots possibly affect the mechanical behavior of thick NiTi layer?

Author Response

We thank to the reviewer for his/her comments. The answers can be found in the attached word file "answers to Referee 1"

Reviewer 2 Report

I have read the manuscript titled "Al2O3 and Pt atomic layer deposition for surface modification of NiTi shape memory films" in detail, and find that there are interesting features here that warrant publication in Coatings. There are however, a range of smaller issues that must be sorted out before it can be published.

  1. Please take time and care to revise the written English. At this stage, there are too many errors.
  2. The description of ALD is oversimplified, and references seem to appear very unmotivated. Please see suggestions for more relevant references.

Furthermore, there are some more detailed issues that I have added to the attached annotated .pdf.

With these issues sorted, I believe this manuscript may be suited for publication in Coatings.

Author Response

We thank to the reviewer for his/her comments. The answers can be found in the attached pdf file "answers to Referee 2"

Reviewer 3 Report

Dear Sirs,

The article entitled “Al2O3 and Pt atomic layer deposition for surface modification of NiTi shape memory films” had the main goal of evaluating the production of uniform, conform and reproducible Al2O3 and Pt coatings onto NiTi shape memory alloy films.

The article is interesting and well-written. Novelty, impact on research community and future perspectives could be better clarified, particularly in the Abstract and Conclusion sections. Some concepts should be further described to be clear to a broader audience. Also, past tense to describe methodology performed and discussing results should be used. Please uniformize letter type throughout the text. Other improvements are suggested. In more detail:

  • Abstract: I would personally star with the features added by Al2O3 and Pt inclusion as coatings of NiTi films in light of prospective applications. Then, indicate ALD and Niti advantages. Novelty should be clear. What did you do in this work that has never been done before? How can that be useful?
  • Introduction: “The authors of Ref. [36] (also Ref. [48]) achieved a thin ALD Pt compact layer by depositing an additional W ALD layer on Al2O3 as an adhesion layer. However, in our study, precursors of W were not available to us.” – I would remove this from the introduction and place it in the discussion section, additionally indicating the reason behind it (economic reasons, break of stock, etc.).
  • Materials and Methods: “A variety of experimental work done on the above mentioned samples is 112 described in Ref. [49].” – what exactly? Please add what did you do to the purchased films before ALD; above mentioned – aforementioned?; “In some Pt ALD cycles, hydrogen gas was used instead of dry air.” – please indicate why?; missing description of SEM abbreviation as well as operating conditions, area and depth analysed by EDS.
  • Results: introduce and add details on data acquisition with the HAADF STEM in the Materials and Methods section.
  • Discussion: 1μm2 – 1 μm2

Author Response

We thank to the reviewer for his/her comments. The answers can be found in the attached pdf file "answers to Referee 3"

Round 2

Reviewer 1 Report

Something is still wrong with the organization of XRD in this paper, whereas/at the same time, the XRD is the main tool appropriate for the recognition of Martensitic temperature shift.

In the manuscript text the authors declare the following „In the present study, the Ms transformation temperatures of samples Pt100 and Pt200 were determined using XRD in order to find out any effect of the Pt and Al2O3 ALD processes applied on NiTi/Si samples.“

In the rebuttal letter, the authors tell in the rebuttal letter the following: „The determination of Ms of the NiTi films using XRD is described in a recent paper [Vokoun et al. Effect of FIB milling on NiTi films and NiTi/Si micro-bridge sensor, Smart Mater and Struct, 2020] (films from the same producer as those used in the present manuscript). Also diffractograms are shown in this cited paper. This is why no diffractograms are shown in the manuscript but reference [Vokoun et al. Effect of FIB milling on NiTi films and NiTi/Si micro-bridge sensor, Smart Mater and Struct, 2020] is given in the manuscript.“

This is actually not entirely satisfactory. Very basically, there is no data visible in this manuscript, supporting the statement that „the Ms transformation temperatures of samples Pt100 and Pt200 were determined using XRD“, whatsoever. There is only a reference to a methodology applied earlier in ref. no. 53 on non-coated sample substrate films, but no data is presented allowing one to decide on any comparatively detectable changes in diffractograms taken from non-coated and coated NiTi films. It is a bit weird, that the authors have omitted the evidently most important original XRD results, which are most relevant to the topics, and preferred (even relatively exaggerate) extensive XPS and AFM.

In spite of the methodological deficiency indicated above, the paper can be published already in the present form. One has to keep in one’s mind, nevertheless, that the quality of the presentation of the experimental results in this paper will remain far from excellent.

Author Response

We thank the reviewer for his/her comments. The missing diffractograms were inserted in the revised (2 round) manuscript (Figure 9). The explaining text for Fig. 9 was also added.

Reviewer 2 Report

I have carefully re-read the revised manuscript by Vokoun et al., and I now find it in a state that is suitable for publication.

I have one small suggestion, and that is to rename the sample that is currently called "Al2O3" without subscripts. I think this is really confusing for the readers. Why not call it "alumina sample" or something, to avoid confusiojn between the sample and physical Al2O3?

Other than that, I am happy with the overall quality of the manuscript at this stage.

Author Response

We thank the reviewer for his/her comments. As suggested, the sample's name was changed to Alu10 instead of Al2O3.